 SHORT REPORT

# Theta- and gamma-band oscillatory uncoupling in the macaque hippocampus

Saman Abbaspoor[1], Ahmed T Hussin[2], Kari L Hoffman[1,2]*

[1]Department of Psychology, Vanderbilt Vision Research Center, Vanderbilt Brain Institute, Vanderbilt University, Nashville, United States; [2]Department of Biology, Center for Vision Research, York University, Toronto, Canada

**Abstract** Nested hippocampal oscillations in the rodent give rise to temporal dynamics that may underlie learning, memory, and decision making. Although theta/gamma coupling in rodent CA1 occurs during exploration and sharp-wave ripples emerge in quiescence, it is less clear that these oscillatory regimes extend to primates. We therefore sought to identify correspondences in frequency bands, nesting, and behavioral coupling of oscillations taken from macaque hippocampus. We found that, in contrast to rodent oscillations, theta and gamma frequency bands in macaque CA1 were segregated by behavioral states. In both stationary and freely moving designs, beta2/gamma (15–70 Hz) had greater power during visual search whereas the theta band (3–10 Hz; peak ~8 Hz) dominated during quiescence and early sleep. Moreover, theta-band amplitude was strongest when beta2/slow gamma (20–35 Hz) amplitude was weakest, instead occurring along with higher frequencies (60–150 Hz). Spike-field coherence was most frequently seen in these three bands (3–10 Hz, 20–35 Hz, and 60–150 Hz); however, the theta-band coherence was largely due to spurious coupling during sharp-wave ripples. Accordingly, no intrinsic theta spiking rhythmicity was apparent. These results support a role for beta2/slow gamma modulation in CA1 during active exploration in the primate that is decoupled from theta oscillations. The apparent difference to the rodent oscillatory canon calls for a shift in focus of frequency when considering the primate hippocampus.

*For correspondence:
kari.hoffman@vanderbilt.edu

**Competing interest:** The authors declare that no competing interests exist.

## Editor's evaluation

The rodent hippocampus is one of the main neuroscience models for memory, navigation, and plasticity, and research has suggested important roles for the theta-rhythmic modulation of firing activity and γ oscillations in these processes. This valuable study shows solid evidence of differences between a non-human primate and the rodent hippocampus, in that theta and γ frequencies are segregated by behavioral states, with theta dominating in quiescence and early sleep and β/γ during visual search.

## Introduction

Hippocampal oscillations are heralded as canonical examples of how oscillations support cognition by coordinating neural circuit dynamics (*Buzsáki and Draguhn, 2004*; *Colgin, 2016*; *Hahn et al., 2019*; *Klausberger and Somogyi, 2008*). In turn, behavioral states constrain and entrain specific neural oscillations. In rodents, locomotion and other exploratory movements elicit an ~8 Hz theta oscillation in hippocampal CA1 (*Buzsáki, 2002*; *Kramis et al., 1975*; *Vanderwolf, 1969*; *Whishaw and Vanderwolf, 1973*) and a faster gamma oscillation (25–100 Hz) that nests within theta (*Bragin et al., 1995b*; *Colgin and Moser, 2010*; *Colgin, 2016*; *Csicsvari et al., 2003*). In contrast, during quiescent states, theta and gamma oscillations are suppressed and sharp-wave ripple (SWR) complexes emerge,

the latter consisting of large high-frequency oscillations (HFO, 150–250 Hz) in CA1 that occur within a slower (sharp-wave) deflection (*Buzsáki, 2015*; *Ylinen et al., 1995*). Although the occurrence of SWRs during quiescence is highly conserved across species (*Buzsáki, 2015*), its dichotomy with theta is questionable (*Hussin et al., 2020*; *Leonard et al., 2015*). This may stem from differences in how and when theta oscillations appear across phylogenetic order (*Green and Arduini, 1954*; *Ulanovsky and Moss, 2007*), particularly among primates, including humans (*Courellis et al., 2019*; *Green and Arduini, 1954*; *Halgren et al., 1978*; *Herweg et al., 2020*; *Jacobs, 2014*; *Mao et al., 2021 Stewart and Fox, 1991*; *Talakoub et al., 2019*; *Tamura et al., 2013*). Consequently, gamma coupling to hippocampal theta (*Bragin et al., 1995a*; *Colgin, 2016*; *Lisman and Jensen, 2013*), and the presence – as postulated – of sub-bands of gamma (*Buzsáki and Wang, 2012*; *Colgin, 2016*; *Colgin et al., 2009*; *Csicsvari et al., 2003*), could understandably be affected by the scarcity of theta oscillations in monkeys during species-relevant exploration (*Courellis et al., 2019*; *Hoffman et al., 2013*; *Jutras et al., 2013*; *Leonard et al., 2015*; *Mao et al., 2021*; *Skaggs et al., 2007*; *Talakoub et al., 2019*). In the present study, we therefore adopted a hypothesis-generating (data-driven) approach to identify (i) which oscillatory bands emerge in macaque hippocampal CA1 as a function of behavioral state; (ii) whether these oscillations coalesce or compete; and (iii) to what extent local single units are modulated at these rhythms.

## Results

### Spectral analysis of hippocampal LFP during active visual search and quiescence

We recorded 42 sessions (M1: 26 sessions, M2: 16 sessions) in the hippocampal CA1 subfield of two macaques (*Figure 1*; *Figure 1—figure supplement 1*) during active visual search and quiescence (henceforth: 'rest'). As a control for the effects of the stationary animal, we recorded from one of the above animals (M2) and a third animal (M3) in freely moving and overnight sleep conditions (*Figure 1—figure supplement 2*). Consistent with previous reports (*Leonard and Hoffman, 2017*; *Leonard et al., 2015*), we observed bouts of roughly 20–30 Hz oscillations predominantly during search, and slower-frequency, larger-amplitude local field potentials (LFPs) during rest (*Figure 1A and C*). To visualize the relationship between spectral power across frequency bands, we sorted quantiles of ~1 s segments based on their average power in the 20–30 Hz frequency band, revealing an antagonistic relationship between 20–30 Hz and <10 Hz frequencies (*Figure 1B*), that is when power at 20–30 Hz was greatest, <10 Hz power was qualitatively weakest. In contrast, stronger power at <10 Hz was accompanied by stronger power at >80 Hz. To determine whether the spectrum varies with behavioral epoch, as it does in rodents (*Buzsáki, 1996*; *Whishaw and Vanderwolf, 1973*), we calculated the power spectrum for each behavioral state (*Figure 1D*). To identify power beyond the 1/$f$ background, we used the aperiodic-adjusted power spectrum (*Demanuele et al., 2007*; *Donoghue et al., 2020*). We found stronger 7–10 Hz power during rest compared to active search (*Figure 1D*, middle). In contrast, power in the higher frequencies from 15 Hz to 70 Hz was stronger during active search compared to rest, with a peak in the 20–30 Hz range (both, $p < 0.05$, Wilcoxon signed rank test with FDR correction). In most non-rodent species examined, including humans, hippocampal theta-band oscillations during alert wakefulness are described as occurring intermittently, in short-lived bouts, unlike the protracted and predictable trains of theta oscillations seen in the rodent hippocampus (*Green and Arduini, 1954*; *Jacobs, 2014*; *Jutras et al., 2013*; *M Aghajan et al., 2017*; *Talakoub et al., 2019*; *Ulanovsky and Moss, 2007*; *Watrous et al., 2013*). To identify oscillations that are rare and short lived, and to allow for more direct comparison to conventions used in human iEEG/macroelectrode studies, we used the BOSC method (*Caplan et al., 2001*; *Hughes et al., 2012*; *M Aghajan et al., 2017*). This method quantifies the fraction of time that band-limited power exceeds amplitude and duration thresholds. The amplitude threshold is set after fitting signal to a log-log linear regression to account for the spectral tilt (1/$f^x$) of the distribution under consideration and accepting only residuals with at least 3 cycles exceeding 95% of the $\chi^2$ distribution. This classifies the graded power spectral measure into discrete 'hits' and 'misses' across time. Consistent with continuous power spectral results, we found that theta bouts ('hits') were more prevalent during rest/sleep compared to online active behavioral states. The pattern was the opposite for beta2/gamma frequencies, which were more prevalent during active states (*Figure 1—figure supplement 2*, $p < 0.05$, Wilcoxon signed rank test with FDR correction).

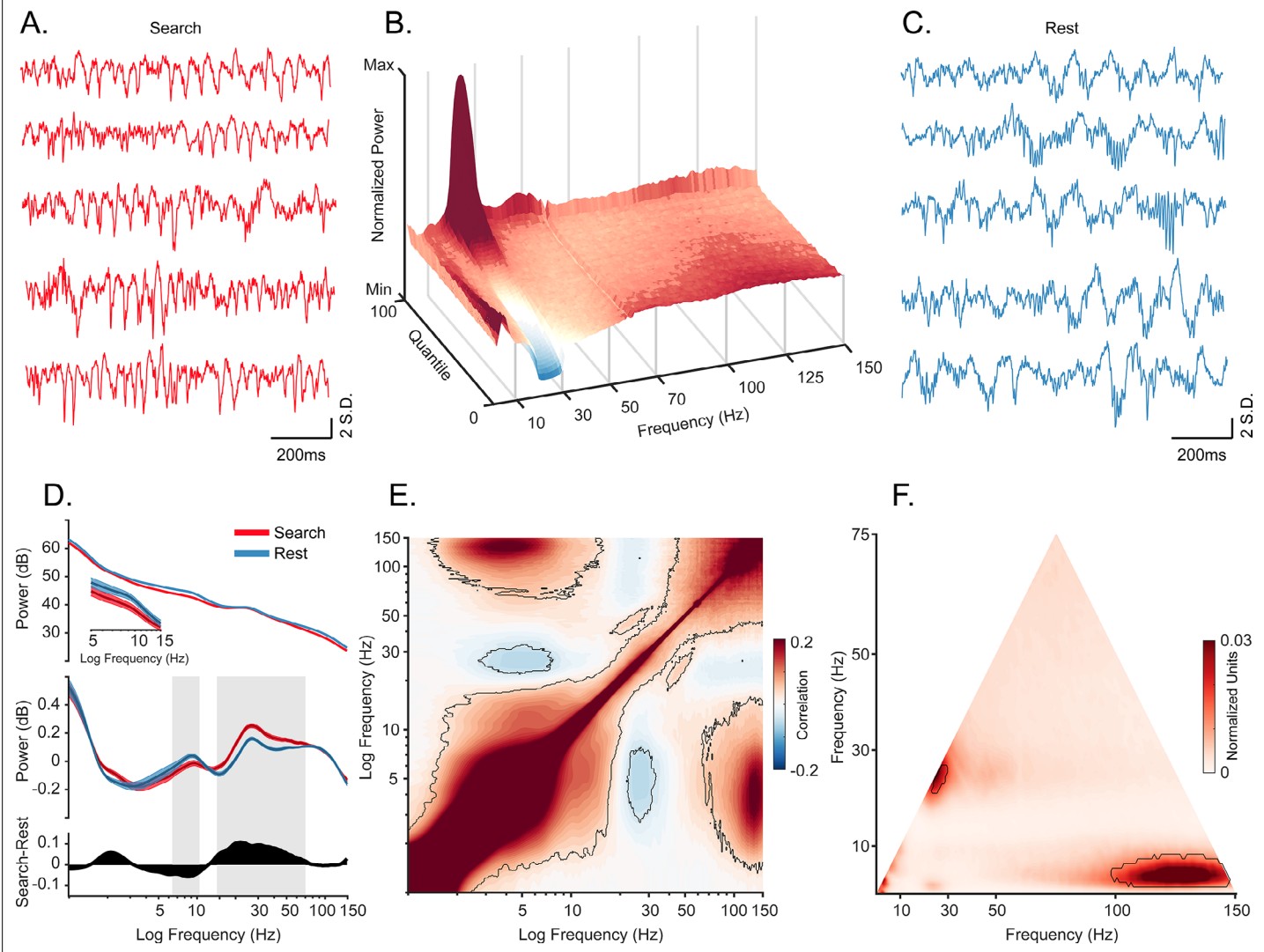

**Figure 1.** Oscillatory decoupling in CA1 field potentials. (**A**) Example traces of broadband local field potential (LFP) in CA1 during search. Data segments were taken from epochs with characteristic high 20–30 Hz power, shown in B (traces were linearly detrended for visualization). (**B**) Spectral density sorted by 20–30 Hz power. Surface plot shows data segments sorted into quantiles according to 20–30 Hz power, revealing an apparent increase in 5–10 Hz power when 20–30 Hz power is weakest. See Methods for details. (**C**) Example traces of wideband LFP in CA1 during the rest epoch showing characteristic interactions between <10 Hz and >60 Hz oscillations. Conventions as in A. (**D**) *Top.* Mean power spectral density during search (red), and rest (blue). *Inset*: mean power for low frequencies of main plot, with shaded 95% bootstrap confidence interval (*N*=42 sessions). *Middle.* Power spectral density after fitting and subtracting the aperiodic 1/*f* component during search and rest, with shaded 95% bootstrap confidence intervals. Gray areas show significant differences in power across behavioral epochs (p<0.05, Wilcoxon signed rank test, FDR corrected) *Bottom.* Power difference between search and rest. (**E**) Average cross-frequency power comodulogram (*N*=42 sessions). Dark outline represents areas that were significant in at least 80% of samples (p<0.05, cluster-based permutation test corrected for multiple comparisons). (**F**) Average bicoherence of the CA1 LFP (*N*=42 sessions). The dark outline represents areas that were significant in at least 80% of sessions (p<0.05, Monte Carlo test corrected for multiple comparisons). The decoupling is preserved when applying analysis methods sensitive to transient oscillations (*Figure 1—figure supplement 2*), and when analyzing CA1 LFPs from an additional monkey who moved freely in a search task, and during night-time rest (*Figure 1—figure supplement 3*).

The online version of this article includes the following figure supplement(s) for figure 1:

**Figure supplement 1.** Electrode localization.

**Figure supplement 2.** Spectral strength and coupling using alternate analysis methods.

**Figure supplement 3.** Oscillatory decoupling in CA1 of freely behaving monkeys.

**Figure supplement 4.** Relative probability distribution of bout durations.

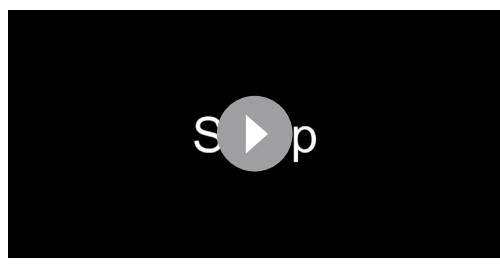

**Video 1.** Oscillatory dynamics in monkey CA1 during sleep and waking states. (Top) Broadband local field potential (LFP) during sleep (blue), and free movement in an enclosure during a search task (red), the 4–8 Hz bandpass filtered LFP, and the 25–50 Hz LFP, shown top to bottom, respectively. Detected bouts in each frequency band are highlighted with blue (sleep) and red (waking). The black vertical line shows 3 SD above the mean. (Bottom) Left: Movement, expressed as the vector norm of angular velocity. The gray horizontal line shows the threshold for movement. Middle: The envelope peak of detected theta (4–8 Hz) bouts plotted as a function of the gamma (25–50 Hz) peaks. Right: 1/f corrected (fitted residual) power spectrum. The distribution of power shifts to higher frequencies during awake compared to sleep.

https://elifesciences.org/articles/86548/figures#video1

In rodents, the frequency and the amplitude of theta oscillation is related to the speed of loco-motion (*Buzsáki, 2002*; *Fuhrmann et al., 2015*; *McFarland et al., 1975*; *Sheremet et al., 2016*) although theta activity is also observed during awake immobile states of alertness (*Kramis et al., 1975*; *Sainsbury, 1998*; *Tai et al., 2012*; *Vanderwolf, 1969*). Theta movement correlates raise the concern that the scarcity of theta during active behaviors in the present study might be attributed to the animals' immobility. To address this concern, we recorded wirelessly from the hippocampal CA1 of the second monkey (M2) and a third adult female macaque (M3) during freely moving active states including immersive visual search and during overnight rest/sleep (M3: 15 sessions, M2: 3 sessions). The results match those of the previous experiment, thereby demonstrating that the decrease in theta-band power and the increase in gamma power during active behaviors generalize, that is they were not merely due to the immobile state of the animals (*Figure 1—figure supplement 3*, *Video 1*). Furthermore, the duration and consistency of theta bouts during early sleep indicate that these methods (recording sites, electrode signal, and states) are capable of detecting theta oscillations, but that they appear during different epochs than those for recordings in rats and mice (*Figure 1—figure supplement 3*, *Video 1*, *Figure 1—figure supplement 4*).

## Hippocampal cross-frequency coupling

To assess the coupling of theta and gamma oscillations at a finer temporal scale, we computed the cross-frequency power correlations across the spectrum. Consistent with the qualitative pattern shown in *Figure 1B*, power at 3–8 Hz and 20–35 Hz were negatively correlated (*Figure 1E*, p<0.05, using a cluster-based permutation test corrected for multiple comparisons). In addition, power in the slower, 3–8 Hz band was positively correlated with that of a much faster, 80–150 Hz band. We next applied a complementary approach, comparing the amplitude envelopes of these two bands (theta and slow gamma) over time, to track the finer temporal structure of power correlations. The results supported the epoched data results (*Figure 1—figure supplement 2*).

To estimate phase-amplitude coupling in the LFP, we performed bicoherence analysis (*Figure 1F*; *Giehl et al., 2021*; *Hyafil, 2015*; *Kovach et al., 2018*), revealing a peak cluster around the 25 Hz frequency range which confirms an interaction between the activity at this frequency and its second harmonic. In addition, the 3–8 Hz band was coupled to high frequencies of 95–150 Hz (p<0.05, cluster-based Monte Carlo statistical test). This is consistent with our cross-frequency amplitude coupling results and indicates that the correlated envelopes in *Figure 1E* are driven by phase-specific coupling of the high frequencies (*Figure 1B*). Nevertheless, our bicoherence results showed no significant phase-amplitude coupling between theta and gamma frequency range.

## Oscillatory modulation of spiking activity

Peaks in spectral power do not necessarily indicate the presence of oscillations in the underlying neural activity (*Buzsáki and Wang, 2012*; *Pesaran et al., 2018*; *Herweg et al., 2020*; *Jones, 2016*). If oscillations are present in the local neural population, regular comodulation between spikes and local field oscillation phases should occur. We measured the spike-field coherence for the whole duration of the sessions by calculating pairwise phase consistency (PPC) (*Vinck et al., 2010*) for well-isolated units (*N*=404). Individual cells phase locked to multiple frequencies (*Figure 2A*; p<0.05, permutation test

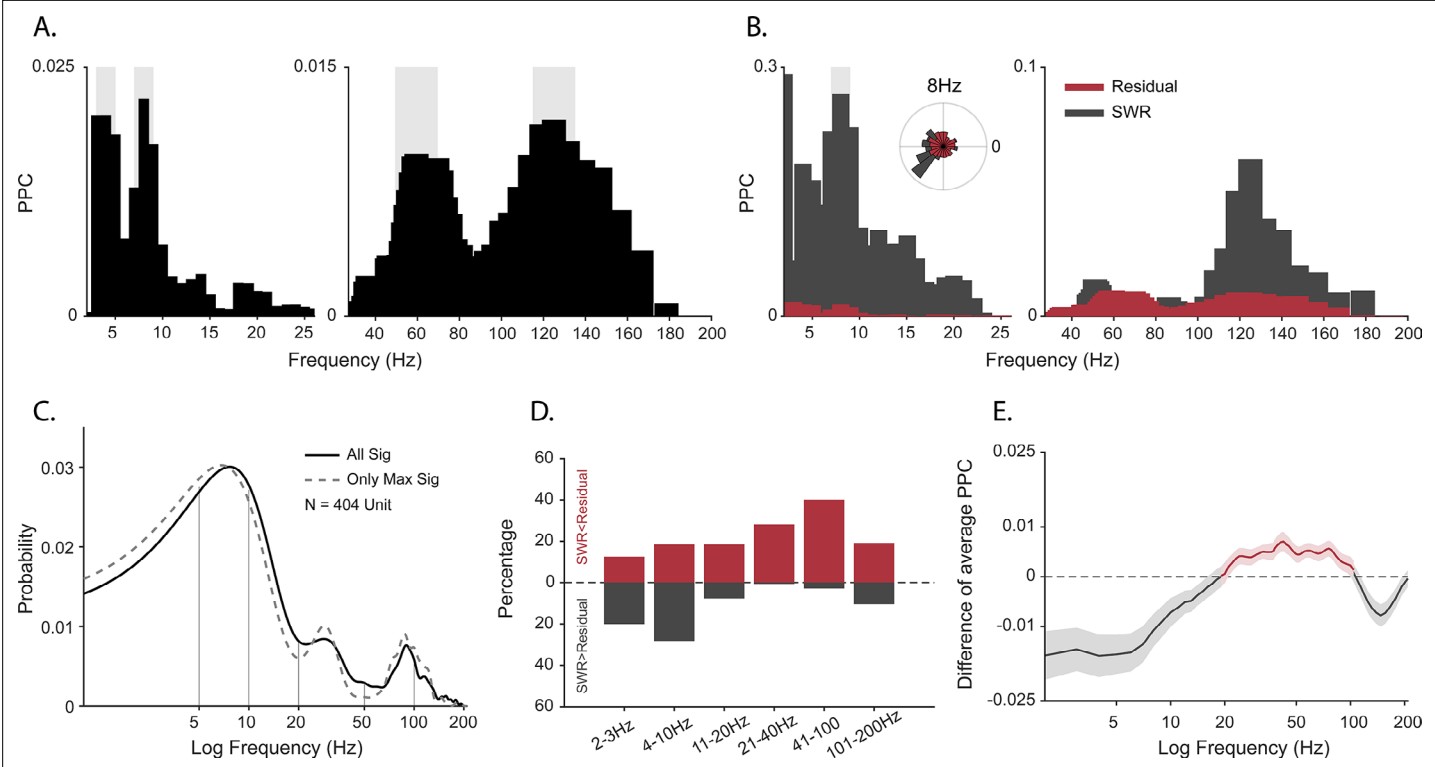

**Figure 2.** Spike-field coherence and its influence by sharp-wave ripples (SWRs). (**A**) Spike-local field potential (LFP) pairwise phase consistency (PPC) spectra for an example unit. Shaded gray shows significant values (p<0.05, permutation test and Rayleigh test, p<0.05). (**B**) Spike-LFP coherence for spikes during detected SWRs (dark gray) and for spikes remaining after extracting the SWR epochs ('residual', red). Light gray shading shows significant difference at p<0.05 in FDR-corrected permutation test for the SWR group. Inset: Normalized histogram of the phase values at 8 Hz, obtained from the spike-LFP coherence analysis shown in A. (**C**) Probability distribution of observing significant PPC values across all frequencies (solid black line), and only for preferred frequency (frequency with maximum PPC value) before adjusting for SWRs (*N*=404 units). (**D**) Difference in the proportion of cells with greater spike-LFP coherence for SWR (gray) and SWR-removed residual (red), for six frequency bands (*N*=185 units). (**E**) SWR residual difference of mean spike-LFP coherence. Shading shows 95% bootstrapped confidence interval (N = 185). Positive values indicating greater PPC for residual than SWR groups are shown in red, negative values (SWR>residual) in dark gray.

The online version of this article includes the following figure supplement(s) for figure 2:

**Figure supplement 1.** Examples of detected ripple events.

**Figure supplement 2.** Spike-train autocorrelograms of example hippocampal cells.

and Rayleigh test, p<0.05), with the population showing the full range of spike preferred frequencies of modulation (*Figure 2C*).

One of the caveats of spike-field coherence measures is that they can be sensitive to large amplitude non-periodic deflections. SWRs have a non-oscillatory amplitude envelope within the frequency range of 0.1–10 Hz in rodents and non-human primates (*Hussin et al., 2020*; *Leonard et al., 2015*; *Maier et al., 2003*; *Rex et al., 2009*; *Skaggs et al., 2007*). Furthermore, we previously observed that SWRs occur in primates during active visual exploration (*Leonard and Hoffman, 2017*; *Leonard et al., 2015*) and that the probability of firing of cells increases during these events, for putative pyramidal and inhibitory cells alike (*Hussin et al., 2020*; *Leonard et al., 2015*; *Skaggs et al., 2007*). Given the limitations of spike-field coherence, the frequency characteristics of SWRs, spiking activity profile of neurons around these events, weaker power at 7–10 Hz during active search, and the strong coupling between bands matching the SWR events (3–8 Hz and >95 Hz), we hypothesized that spike-LFP coherence at low frequencies might be partly produced as a byproduct of the slow deflection of SWRs (*Hussin et al., 2020*) rather than via harmonic oscillations. To test this, we extracted peri-SWR spikes, computed PPC only for these spikes in each cell [PPC$_{swr}$], and then compared this to the PPC for spikes outside the SWR windows [PPC$_{residual}$]. *Figure 2B* shows an example cell that exhibits stronger spike-LFP coherence at 8 Hz during SWR than outside the SWR window (p<0.05, permutation test

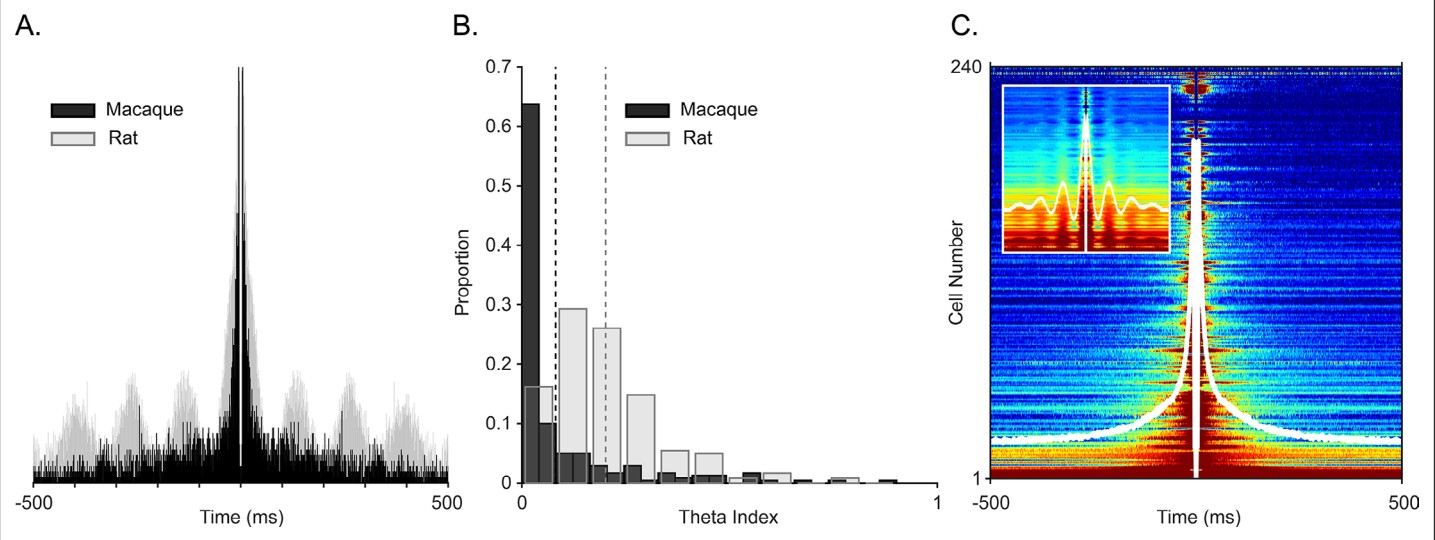

**Figure 3.** Examining spiking periodicity for theta modulation in macaque hippocampus. (**A**) Autocorrelogram of an example 'theta' unit from *Figure 2A* (black, N=945 total spikes) and a theta-modulated unit in rat (gray, N=5655 total spikes). (**B**) Distribution of the theta index in CA1 units from this study (black, N=240 units), and CA1 units of rat (gray, N=197 units). Dashed lines show mean values, color coded by species. (**C**) Sorted autocorrelograms of CA1 units, with mean, population ACG shown as a white trace. Inset: same as the main plot but for rats.

with FDR correction). At the population level, coherence at lower frequencies (2–10 Hz) was greater during SWR than in the SWR-removed distributions. In contrast, the higher frequencies (10–200 Hz) maintained a greater coherence outside of the SWR time window (*Figure 2D*). This led to weaker mean spike-field coherence restricted to the <10 Hz range, after removing the influence of SWRs, suggesting a contribution of the non-oscillatory slow deflections in the SWR complex to the apparent cross-frequency interactions.

To ensure that oscillations were local and to avoid the influence of aperiodic deflections, we generated spike autocorrelograms. Periodic peaks of the spike autocorrelograms demonstrate theta rhythmicity in rat and mouse CA1 (*Cacucci et al., 2004*; *O'Keefe and Recce, 1993*; *Royer et al., 2010*). To directly compare theta rhythmicity in our cell population with observations in the rodent, we computed the autocorrelogram for a complete population of well-isolated cells with at least 100 spikes for the whole session and for the complete cell population from homologous subregions of CA1 in rodents, also across the whole session. *Figure 3A* shows an autocorrelogram of the theta spike-field coupled cell in *Figure 2A* (black, monkey) overlaid onto a theta rhythmic cell from the homologous CA1 region of a rat (i.e. temporal CA1) shown in gray. To quantify theta autocorrelogram rhythmicity, we calculated the theta modulation index (*Jacobs, 2014*). Compared to modulations seen in the rat (*Figure 3B* in gray), monkey cells typically showed near zero index values, that is, they were not modulated (*Figure 3B* in black). This was also evident in the sorted-cell and mean population autocorrelogram (*Figure 3C*, inset shows rat distribution). For a given firing rate, higher frequencies are less likely to demonstrate cycle-by-cycle periodicity due to their shorter periods; nevertheless, we observed a few examples of >20 Hz spike modulation in the spike-triggered averages and autocorrelograms of several cells (*Figure 4*).

## Discussion

In this study we evaluated the oscillatory dynamics of primate hippocampus during two general behavioral states: first, during 'online' states of awake active behavior and second, during offline states. The waking-behavior recordings included stationary monkeys who were engaged in hippocampal-dependent, memory-guided visual exploration (*Chau et al., 2011*; *Dragan et al., 2017*; *Yoo et al., 2020*), and freely moving monkeys exploring their environment. The offline-state recordings included post-task quiescent states in the darkened booth, and the early stages of overnight sleep, respectively. The most prominent oscillation in amplitude and prevalence during alert active behavior was in the beta2/slow gamma band (~20–35 Hz for stationary search and ~20–50 Hz in freely moving subjects).

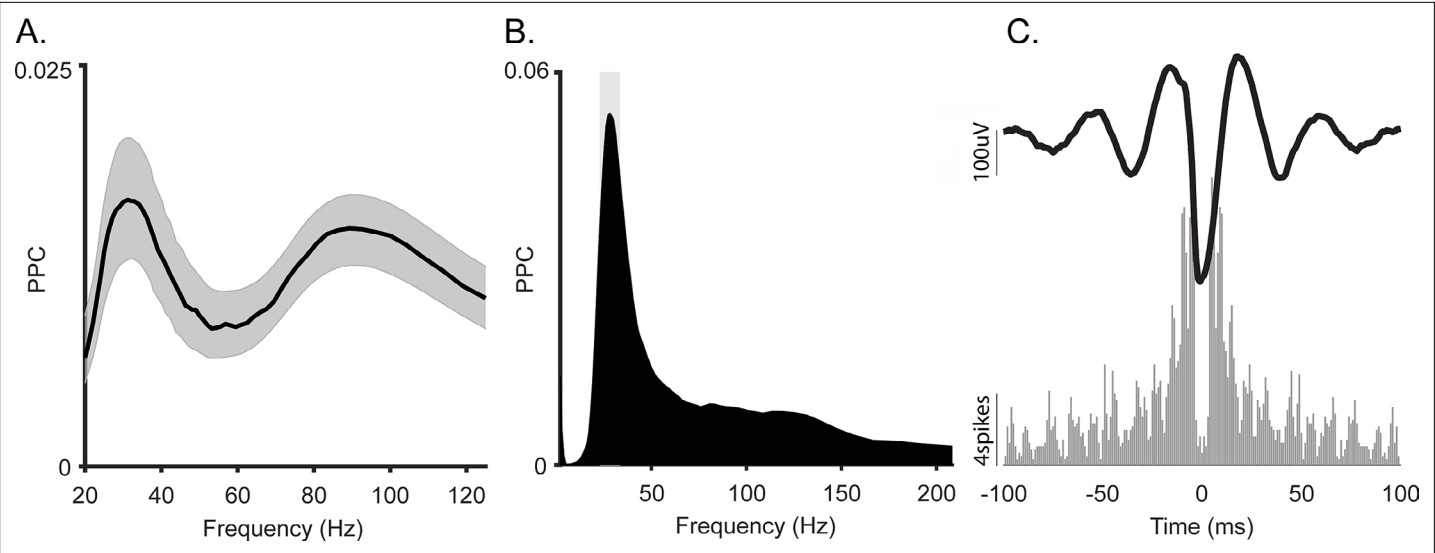

**Figure 4.** Spike-local field potential (LFP) coherence of the slow gamma oscillation in macaque hippocampus. (**A**) Average spike-LFP coherence in gamma frequency range. Shading shows 95% bootstrap confidence interval (N = 404). (**B**) Spike-LFP coherence for a representative gamma-locked unit. This unit had a significant peak at 30 Hz (p<0.05, permutation test and Rayleigh test, p<0.05). (**C**) Top: Spike-triggered average LFP of the unit in B. Bottom: Autocorrelogram of the same unit.

In contrast, theta-band (3–10 Hz, peak at ~8 Hz) amplitude and bout prevalence were greater during offline states of rest and sleep than during online active states. This bimodal oscillatory profile was also evident in finer temporal scales. Theta and beta2/gamma bands were decorrelated in the cross-frequency amplitude coupling, and in phase-amplitude bicoherence, quite unlike that observed in rodents (*Zhou et al., 2019*). Spike modulation by LFP frequency was seen in both bands; however, only the theta-band effect appeared to be partially due to non-oscillatory field events, such as the SWR. Despite longer and more prevalent theta oscillations in rest, overall, we found no clear evidence of theta-periodic modulation. This contrasts with the strong and prevalent modulation seen using the same analyses methods applied to signal from the rat in homologous regions of CA1. Across several measures we found consistent or compatible results in the coalescence among frequency bands, and between bands and behavioral states within this study. These patterns differ from well-established spectral-behavioral coupling in rodents.

## Dormant theta

In rat and mouse hippocampal LFPs, theta activity (centered at 6–9 Hz) is most consistently present during activated cortical states, that is, during alert aroused behaviors and REM sleep (*Buzsáki, 2002*; *Colgin, 2013*; *Nuñez and Buño, 2021*; *Vanderwolf, 1969*). In contrast, theta gives way to SWRs and irregular slow activity during quiescence and NREM sleep (*Buzsáki, 2015*; *Buzsáki, 1989*). Further, theta 'nests' different frequencies of gamma in a phase-specific manner during these 'active' cortical states (*Colgin and Moser, 2010*; *Colgin, 2016*; *Lasztóczi and Klausberger, 2014*; *Lisman and Jensen, 2013*). And finally, most classes of cells in CA1 are modulated by the theta rhythm (*Klausberger and Somogyi, 2008*; *Klausberger et al., 2003*; *Royer et al., 2010*). Indeed, theta is so reliably present and so spectrally dominant in fields, spikes, and intracellular currents during the active states when gamma occurs, that studies of hippocampal gamma oscillations and spikes are commonly conditioned on them.

Monkey CA1 LFPs also show reliable, durable theta oscillations with peak frequencies at 6–9 Hz; however, where measured across states, they do not demonstrate the above characteristics of theta. Instead, they show divergent brain-behavior coupling to that seen in rodents. Much of the literature in humans and monkeys has focused on brief epochs of alert, goal-directed behaviors within stationary visual and visuospatial tasks and may therefore be suboptimal for detecting theta oscillations. Yet where rest or sleep have been measured, stronger, more durable theta field potential oscillations appear during quiescence and NREM sleep (or anesthetized states, *Kleen et al., 2021*) than during

alert/task-on states or REM sleep (*Bódizs et al., 2001*; *Cox et al., 2019*; *Moroni et al., 2007*; *Tamura et al., 2013*; *Uchida et al., 2001*). Whereas these behavioral-state correlates of field potential oscillations apparently differ by species, local control of unit activity may nevertheless reveal strong underlying rhythmic motifs.

To better understand local modulation, we measured CA1 spiking activity as a function of LFP oscillatory phase. All tested frequencies had at least some units that were preferentially phase locked to them, but the most common preferred frequencies were roughly matching the 3–10 Hz, 20–30 Hz, and 60–150 Hz bands. The lowest band would be consistent with theta modulation of local spiking, which has been reported using a few different methods in human and macaques. Spike-field coherence in the theta frequency range was seen for microelectrode bundles including microwires in unspecified subregions of the human hippocampus (*Rutishauser et al., 2010*), and for recording sites estimated to be within the hippocampus proper of the freely moving macaque (*Mao et al., 2021*), though in the latter case, the higher gamma bands and delta bands were several times more likely to show SFC. Within the theta band, spurious coherence may arise in response to nonrhythmic deflections (*Aru et al., 2015*; *de Cheveigné and Nelken, 2019*; *Jones, 2016*; *Vinck et al., 2022*), such as the slow components of the SWR which is especially prominent in macaques and human (*Hussin et al., 2020*; *Liu et al., 2022*). Indeed, we found that SWRs (*Figure 2—figure supplement 1*) disproportionally influenced the 2–10 Hz band (*Figure 2B, D and E*), whereas the gamma frequency range was largely unaffected. Because SWRs occur infrequently, it was somewhat surprising that SWR removal affected the overall results; however, the population activity during ripples typically includes 2–10× firing rate increases in nearly all cell types (*Buzsáki, 2015*; *Hussin et al., 2020*; *Leonard et al., 2015*; *Skaggs et al., 2007*). As such, each ripple is likely contributing disproportionately to the spike-locked activity. In addition, we know that visual (*Katz et al., 2020*; *Rey et al., 2014*; *Roux et al., 2022*) and saccade-elicited ('ERP-like') responses in the hippocampus (*Hoffman et al., 2013*; *Jutras et al., 2009*; *Katz et al., 2022*) increase theta-band power without necessarily producing harmonic oscillations. This would presumably only affect the segments from search, not sleep. Future human and monkey hippocampal studies should factor out the contributions of SWRs, and to the same end stimulus and saccade-locked transients, when evaluating low-frequency (<10 Hz) oscillations for non-stationary signals and when using methods that do not discriminate oscillations from transient deflections, as with spike-field coherence. The present results found residual coherence, and in addition, the typical duration of sleep theta bouts would well exceed the SWR envelope (as shown in *Video 1 and Figure 1—figure supplement 4*), therefore we consider the observed spectral peaks during rest likely reflect true oscillations and not simply collections of evoked transients.

The spike autocorrelogram is a more stringent measure of oscillatory modulation, not susceptible to LFP spectral analysis artifacts. The <10 Hz range including the theta band failed to show clear examples of theta modulation (*Figure 2—figure supplement 2*). This was not simply due to low firing rates: even high firing rate cells, which should include 'theta-on' interneurons, failed to show theta modulation. Indeed, the same analysis applied to units from a homologous region of rat CA1 (*Royer et al., 2010*) showed a predominance of theta modulation (even after pooling functional cell types). In light of our rest-related theta power and SWR-removed spike-field coherence, this was somewhat surprising, but consistent with the results in free-flying bats (*Yartsev et al., 2011*), monkeys (*Courellis et al., 2019*; *Mao et al., 2021*), and humans (*Qasim et al., 2021*). In such cases, to observe spike timing regularity to field potentials, the aperiodic – that is non oscillatory – slow fluctuations in the LFP must first be warped to 'fit' one another, to create a pseudo-oscillation (*Bush and Burgess, 2020*; *Eliav et al., 2018*; *Mao et al., 2021*; *Qasim et al., 2021*). Due to the different criteria used, compared to the theta rhythmic spiking reported in rodents, the theoretical or computational roles ascribed to the theta oscillations may need to be re-evaluated, or, more precisely defined in terms that do not depend on periodicity.

Slower oscillations, or irregular low-frequency components at 1–4 Hz 'delta' band, are more frequently correlated with spatial or associative memory effects (*Goyal et al., 2020*; *Vivekananda et al., 2021*; *Watrous et al., 2013*), leading to the supposition that rodent hippocampal theta is simply faster than in primates (*Jacobs, 2014*; *Mao et al., 2021*). If our present results were based on two distinct oscillations – the faster 'offline' theta and the slower 'online/memory' delta band – we would have predicted anti-correlations or segmentation in the power plots into sub-bands. We have yet to see such a segmentation, but we would stress that this remains a candidate for closer inspection. Finer

task behaviors might help to uncover this possible alert-behavior correlate. Even if this proves to be the case, it would raise further questions about which bands become considered homologous across species. If 1–4 Hz is envisaged as relating to septal-cholinergic theta in rats, what is the equivalent of the classic 8 Hz theta band in rats? Compounding this problem is a separate 1–4 Hz oscillation that was observed in rat hippocampus (*Jackson et al., 2014*; *Schultheiss et al., 2020*). When trying to consider homology across species, the LFP signals are only proxies for the underlying circuit activity. Identifying the local circuit motif, including receptor-specific neuromodulation in primates (*Stewart and Fox, 1991*), may help differentiate among various low-frequency oscillations and their functional roles across species.

## Decoupled gamma

Gamma oscillations in rat and mouse CA1 comprise a wide high-frequency band (initially 40–100 Hz but more recently 20 Hz or 30–100 Hz) known for tight coupling within the theta oscillation (*Bragin et al., 1995a*; *Buzsáki et al., 1983*; *Colgin and Moser, 2010*; *Colgin et al., 2009*; *Tort et al., 2008*). In line with this coupling, gamma shares behavioral state correlates with theta: waking exploratory states and REM sleep. We found that the slower part of the gamma range (<~70 Hz) is decoupled from theta. Consistent with this finding, gamma power is seen as a relatively strong oscillation in monkeys and humans during REM sleep (*Cantero et al., 2003*; *Takeuchi et al., 2015*; *Tamura et al., 2013*; *Uchida et al., 2001*), as in rodents. Gamma oscillations, when measured, are often associated with hippocampal processing (e.g. memory guided search, *Leonard et al., 2015*; *Montefusco-Siegmund et al., 2017*), and retrieval (*Montefusco-Siegmund et al., 2017*) or subsequent memory effects (*Jutras et al., 2009*) in monkeys, or spatial coding and memory in rodents. In addition, a band-limited 20–40 Hz oscillation is seen in rodent CA1 (and is synchronized with LEC, *Igarashi et al., 2014*) during discrete item cueing or retrieval such as in olfactory associative place and sequence learning (*Allen et al., 2016*; *França et al., 2014*; *Lansink et al., 2016*; *Lopes-Dos-Santos et al., 2018*), and when exploring novel environments (*França et al., 2021*; *Trimper et al., 2017*). This frequency band is associated with activation of the trisynaptic pathway (DG and CA3, *Colgin, 2016*; *Fernández-Ruiz et al., 2021*; *Hsiao et al., 2016*; *Rangel et al., 2015*), but is also a coherent oscillation between CA1 and LEC (*Igarashi et al., 2014*) implicating direct, temporoammonic pathways. Finally, beta2/gamma coherence is seen in hippocampal-retrosplenial and hippocampal-mFPC oscillations in monkeys during object-scene associative memory (*Hussin et al., 2022*). The range of implicated pathways and regions underscores the need to identify the underlying microcircuits that give rise to these oscillations (*Fernandez-Ruiz et al., 2023*). Determining which are conserved across species will be an important topic for future research.

## Gamma and theta coupling

The results of the current study show that theta oscillations are not prominent during awake behavior and do not couple with gamma, and thus, are not a good candidate for structuring possible gamma sub-bands. Although most studies in humans focus on coupling via slower or faster bands outside this range, we note two exceptions that ostensibly find increases in hippocampal theta-gamma coupling associated with active processing in the theta band (*Axmacher et al., 2010*; *Stangl et al., 2021*) and a third that included 5–11 Hz, for which the low-frequency-granting signal may have included the alpha band (*Roux et al., 2022*). Both of the former studies use macro-electrode iEEG with >1 mm contacts and 3–10 mm spacing between contacts, along with MR/CT coregistration, which are estimated to be in some regions of the hippocampus proper in only a subset of participants. Thus, any apparent discrepancies to the present findings may involve the inclusion of extra-hippocampal signals, though this would still be interesting to understand. Signal localization notwithstanding, the first study of working memory reported 6–10 Hz medial temporal lobe theta phase that was coupled with gamma oscillations (*Axmacher et al., 2010*). Although modulation strength was not associated with performance, modulation width predicted reaction time. Pertinent to the state correlates of theta, they show that the intertrial interval (ITI) – that is a potentially offline state – shared a strong modulation, similar to that seen during the memory epoch, except at the onset of the WM epoch, where intertrial coherence was strongest. In the second study, recording MTL iEEG in ambulating patients, theta power did not increase with movement speed (if anything, it nominally decreased), and search during movement was associated with weaker theta; however, when stationary, some theta modulation was

seen and in addition, general theta-gamma coupling was reported between 6 Hz and 10 Hz and high gamma during movement and between 6 Hz and 10 Hz and a slower gamma when stationary and observing others (*Stangl et al., 2021*). The effects of saccade-evoked responses on this theta were not described but saccades were associated wtih increases in reported theta power, thus, it remains unclear what effect saccadic responses may have had on the reported theta gamma coupling. In the third study, using microwires, the most applicable measures to our study produced generally consistent results, including decreased theta and increased gamma power associated with successful associative memory, and no difference in low-frequency spike-field coherence for locally measured fields and spikes (*Roux et al., 2022*). Here, unlike the first study (*Axmacher et al., 2010*), the modulation of gamma power by peak 5–11 Hz phase was greater with successful memory (hits vs. misses). This might suggest a theta or alpha oscillation is regulating gamma magnitude; however, the MI measure of phase-amplitude coupling does not require strong or periodic phase-granting signal to cluster gamma (*Tort et al., 2010*). Future studies may help to disentangle the role of periodic theta in contrast to saccadic or other evoked response waves, in clustering active gamma oscillations during active behaviors. At present, we suggest that gamma oscillations – across species – can nevertheless work as a standalone rhythmic activity to select among inputs by virtue of laminar specificity, including activity of other hippocampal fields (e.g. CA3-dentate gyrus) or extrahippocampal areas (e.g. entorhinal cortex).

Our findings offer a hypothesis-generating framework for future analysis in (human and non-human) primate hippocampal physiology. The present results suggest that theta oscillations were not prevalent during search in primates and did not consistently modulate single unit activity, but rather form the strongest oscillatory marker of offline or quiescent states. Instead, beta2/slow gamma oscillations constitute the chief, self-contained oscillation that arises during active exploration in the primate hippocampus and stands as the most likely oscillation for organizing local dynamics during exploration. Aside from understanding the nature of these cross-species differences, future work may focus on the better-conserved aspects of CA1 activity, including gamma synchrony during exploration, locking of spiking to exploratory movements, and aperiodic spike timing measures that don't require autocoherent oscillations. Despite several surface differences in hippocampal-behavioral coupling across phylogenetic orders, the underlying neural circuit activity giving rise to these oscillations may yet reveal fundamentally conserved mechanisms.

## Methods

### Subjects and task

Two adult female macaques (*Macaca mulatta*, 'M1, M2') were used in the main, visuospatial search experiments whose results are shown in all figures except those of the control task/recordings from *Figure 1—figure supplement 3*. Data from M1 and M2 have been reported previously (*Hussin et al., 2020*; *Leonard and Hoffman, 2017*; *Leonard et al., 2015*). The apparatus, training procedure, and task have been described previously (*Leonard and Hoffman, 2017*; *Leonard et al., 2015*), and are summarized briefly here. During search, animals performed a hippocampally dependent visual target detection task. In the task, seated, head-fixed monkeys were placed in front of a monitor and were required to identify a target object from nontargets in unique visual scenes presented on a monitor positioned in front of them, and report their selection of scene-unique target objects by holding their gaze on the target region for a prolonged (≥800 ms) duration. Target objects were defined as a changing item in a natural scene image, where the original and changed images were presented in alternation, each lasting 500 ms, with a brief gray screen (50 ms) shown between image presentations. An ITI of 2–20 s followed each trial. The daily sessions began and ended with a period of at least 10 min when no stimulus was presented within the darkened booth and animals could sleep or sit quietly ('rest'). All procedures associated with this task were conducted with approval from the local ethics and animal care authorities (Animal Care Committee, Canadian Council on Animal Care). Two adult female monkeys (*M. mulatta*, 'M2' and 'M3') were used in a separate experiment to extend the analysis to include freely behaving conditions. The search epochs of M2 were observational (identified post hoc) and of M3 were experimentally controlled. M2 was placed in an enriched environment where she could actively forage, play with toys (manipulanda), walk, climb, self-groom, and groom another animal. Blind raters denoted the times of foraging, walking, and exploratory 'search' behaviors. M3 was placed in a testing enclosure equipped with multiple touchscreens around the periphery

that presented spatially distributed arrays of objects. To obtain fluid reward the monkey was required to locate and select (touch) designated objects in a global spatial sequence across the enclosure, thereby requiring visual search, reaches, and walking/climbing during a trial. For M3 search data analysis, we extracted the trial sequence duration+2 s beyond the first/last touches of the sequence which included goal-directed walking, but excluded the ITI containing reward consumption or idle time prior to the animal's approach and trial initiation. For both M2 and M3, rest epochs consisted of 40 min of recordings in the housing area during the start of the 'night' cycle of the room's automatic lighting system. A total of 18 sessions (M3: 15, M2: 3) were analyzed for both task and rest epochs. All procedures for M3 were conducted in accordance with the approved protocols and authorized procedures under the local animal care authorities (Institutional Animal Care and Use Committee).

## Electrophysiological and movement recordings

For monkeys M1 and M2, indwelling bundles of movable platinum/tungsten multicore tetrodes (96 μm outer diameter; Thomas Recordings) were implanted into the anterior half of hippocampus and lowered into CA1. In M3 we recorded from an indwelling active multichannel probe on an adjustable microdrive ('Deep Array' probe, beta-test design, Diagnostic Biochips, Inc). Recording sites were verified with postoperative CT coregistered to pre-operative MRI and using functional landmarks that changed with lowering depth, including the emergence of depth-specific SWRs in a unit-dense layer, as described in the previous studies (*Figure 1—figure supplement 1*, *Figure 2—figure supplement 1*). Post-explant MRI verified the electrode locations in M1 (*Leonard et al., 2015*). For the current study, we detected channels within the pyramidal layer based on the strongest amplitude of ripples during SWRs and single unit activity, and only used these channels for all analyses. LFPs were digitally sampled at 32 kHz using a Digital Lynx acquisition system (Neuralynx, Inc) and filtered between 0.5 Hz and 2 kHz for M1 and M2. LFPs for M2 and M3 that were used for results shown in *Figure 1—figure supplement 2* were sampled at 30 kHz using the active headstage and recorded using a Cube/Freelynx wireless recording system (Neuralynx, Inc) to SD card (for rest) or using wireless transmission to the Cheetah acquisition system (for task). Single-unit activity was filtered between 600 Hz and 6 kHz, recording the waveform for 1 ms around a threshold-triggered spike events. Spike sorting was performed semi-automatically using KlustaKwik based on wave shape, principal components, energy, and peak/valley across channels. This was followed by manual curation of clusters in MClust (A.D. Redish). We used 3D accelerometer in Freelynx data acquisition system to record the movement of freely behaving animal subjects during active search and rest. Angular velocity (AV) traces were recorded at 3 kHz and samples were synchronized with the neural recording. For movement detection, we first calculated the vector norm of the three AV axes using vecnorm function in MATLAB. Then, we thresholded the vector norm to find periods of time when animal was moving. Threshold was calculated based on the vector norm of AV recordings when Freelynx was placed statically on a flat surface. In addition to the accelerometer data, we used recorded videos to detect animal behavioral state (e.g. sleep).

## SWR detection

SWR detection was performed on the tetrode channels of M1 and M2 with the most visibly apparent ripple activity using the previously described method (*Leonard et al., 2015*). Raw LFPs recorded from the tetrode channel were filtered between 100 Hz and 250 Hz. To determine the SWR envelope, filtered LFPs were transformed into z-scores and rectified and subjected to a secondary bandpass filter between 1 Hz and 20 Hz. Events with a minimum amplitude exceeding 3 SDs above the mean with a minimum duration of 50 ms, beginning and ending at 1 SD were designated as potential ripples. High-frequency energy is present for non-SWR events such as EMG and other non-biological noise, though these artifacts are distinct from ripples because the latter are restricted to the regions near the pyramidal layer. For artifact (non-ripple-event) rejection, a distant tetrode channel was selected as a 'noise detecting' channel (*Talakoub et al., 2016*). Events that were concurrently detected on the noise channel and the 'ripple-layer' channel were removed from the ripple pool. High gamma (80–120 Hz) and HFO (110–160 Hz) events were similarly identified, but with the filter criterion set at 80–120 Hz and 110–160 Hz, respectively, and in both cases identifying peaks as those >1 SD. Duplicate high gamma/HFO and SWR events were labeled as SWRs. Events with a repetition rate <125 ms were considered a single event.

## Power spectral parametrization and fitting

To compare the spectral content during search and rest, we selected successfully completed trials lasting longer than 1 s. For rest segments in the stationary (tethered) recordings M1 and M2, we extracted the LFP signals that were recorded before the start of the task and after the end of the task when the animal was in a dark environment in a quiescent or inactive state. For rest segments of wireless recordings in M2 and M3, we extracted LFP signals recorded during the evening after the task, during the dark cycle of the housing area. We used Welch's method with a 50%-overlapping 1024-sample sliding Hanning window to estimate power spectra for the frequency range of 1–150 Hz with a frequency resolution of 0.25 Hz.

To identify spectral peaks and compare between search and rest states, we parameterized power spectra using the method described by *Donoghue et al., 2020*. This method models power spectra as a combination of the 1/*f* frequency components (aperiodic) in addition to a series of Gaussians that capture the presence of peaks (periodic components). The model was fit to a frequency range between 1 Hz and 200 Hz with a frequency resolution of 0.5 Hz. Settings for the algorithm were set as: peak width limits: (0.5, 12); max number of peaks: infinite; minimum peak height: 0; peak threshold: 2.0; and aperiodic mode: 'Fixed'.

To assess statistical significance for the difference in parametrized spectra at each frequency, we used Wilcoxon signed rank test at $p<0.05$ with FDR correction for multiple comparisons.

## Twenty to 30 Hz sorted spectral density map

We estimated the power spectral density using Welch's method described in the previous section to obtain (frequency * PSD segments) matrix. We then sorted the PSD segments based on the mean power in 20–30 Hz frequency range and normalized each segment by dividing it by its median. We clustered all sorted segments into 50 total segments of equal size (frequency * 50 segments) by averaging original PSD sorted segments. We repeated this procedure across sessions and animals separately.

## Cross-frequency power correlation

On continuous LFP time-series data, Welch's method with a 50%-overlapping 1024-sample sliding Hanning window was used to estimate the spectrogram for the frequency range of 1–150 Hz with a frequency resolution of 0.25 Hz.

We computed the pairwise correlation between cross-frequency power using the following formula (*Masimore et al., 2004*):

$$corr_{ij} = \frac{\sum_k \left(S_k(f_i) - \overline{S}(f_i)\right)\left(S_k(f_j) - \overline{S}(f_j)\right)}{\sigma_i \sigma_j}$$

where $S_k(f_i)$ is the PSD at the frequency $f_i$ in time window $k$, $\overline{S}(f_i)$ the averaged PSD at the frequency $f_i$ over all sliding windows, $\sigma_i$ the standard deviation of the PSD at the frequency $f_i$, and $k$ ranges over all sliding windows. In the wireless recordings in M2 and M3 this procedure was applied to the data segments extracted during task performance.

To test the null hypothesis that the power spectral time series of two different frequencies, $f_i$ and $f_j$, are not coupled, we performed a non-parametric surrogate data method with cluster-based multiple comparison correction (*Thammasan and Miyakoshi, 2020*). This method preserves the original data's statistical properties while generating time series that are randomized such that any possible nonlinear coupling is removed. In this method, we randomized time window $k$ differently for each frequency bin to build surrogate time-frequency time series and computed the surrogate cross-frequency power correlation. This process was repeated 5000 times to produce distributions for the dataset in which the null hypothesis holds. The original, non-permuted data are then compared to the surrogate distribution to obtain uncorrected p-values. The significance threshold was selected to be 0.05. For cluster-based multiple comparison correction, all samples were selected whose p-value was smaller than 0.05. Selected samples were then clustered in connected sets based on their adjacency and the cluster size was calculated. This procedure was performed 5000 times to produce the distribution of cluster sizes. If cluster sizes in the original correlation matrix were larger than the cluster

threshold at 95th quantile, they were reported as significant. We performed this statistical procedure at the level of single channels per animal. We consider as robust significance those areas that were significant for at least 80% of all samples.

## Hilbert amplitude envelope correlation

We applied a third-order Butterworth filter at each frequency of the LFP with a 2 Hz bandwidth. We then Hilbert transformed the bandpass-filtered LFP and estimated the continuous amplitude envelope. We computed the pairwise correlation between cross-frequency amplitude envelope using previously described correlation method.

## Bicoherence

For bicoherence, we used the HOSA toolbox. Bicoherence was estimated for frequencies $f_1$(1–75 Hz) and $f_2$(1–150 Hz) in steps of 1 Hz according to the following formula (*Bullock et al., 1997*):

$$B(f_1,f_2) = \frac{\left| \langle F_t(f_1)F_t(f_2)F_t^*(f_1+f_2)\rangle_t \right|}{\langle \left| F_t(f_1)F_t(f_2)F_t^*(f_1+f_2) \right| \rangle_t}$$

where $F_t(f)$ is the signal's time-frequency transformation at time $t$, || represents the absolute value, and $\langle \rangle$ is the average over time. We set the segment length to 1024 samples for this analysis.

Bicoherence has a higher spectral resolution for disentangling harmonic from non-harmonic cross-frequency coupling. Additionally, bicoherence relaxes the artificial spectral constraints introduced by conventional PAC, corrects for its poor biases, and accounts for asymmetry in the rhythms (*Giehl et al., 2021*; *Kovach et al., 2018*; *Sheremet et al., 2016*).

Theoretically, the bispectrum is statistically zero for linear systems with mutually independent Fourier coefficients. For nonlinear systems, the bispectrum will exhibit peaks at triads ($f_n$, $f_m$, $f_{n+m}$) that are phase correlated, measuring the degree of three-wave coupling (*Sheremet et al., 2016*). In practice, however, bicoherence has a positive bias. The background activity of LFP signals can be estimated by properties of red noise which can then be used for significance testing (*Bédard and Destexhe, 2009*; *Torrence and Compo, 1998*). To calculate the statistical significance of the local autobicoherence, we generated red noise with the same length of our original signals and computed bicoherence for the red noise sample. We repeated this procedure 5000 times to obtain the null distribution. We then compared the original data to the null distribution to obtain uncorrected p-values, thresholded for significance at 0.05. We then performed cluster-based multiple comparison correction as described in the Cross-frequency power correlation section.

## Detection and prevalence of transient oscillatory events

We used the BOSC algorithm (*Caplan et al., 2001*; *Hughes et al., 2012*) to detect transient bouts of heightened frequency-specific power using a joint amplitude and duration thresholding procedure. A specific concern we wished to address was that the 1 s windowed power spectral method may result in false negatives (i.e. missing brief theta epochs), therefore we set a maximally permissive duration threshold of 3 cycles, and an amplitude threshold using sixth-order wavelets passing the 95th percentile of model fit distributions. From these detected events, we computed occupancy rate using the formula:

$$\text{Occupancy rate } (\%) = \frac{\text{Total duration of detected events}}{\text{Duration of the original signal}} \times 100$$

Occupancy rate is a measure of prevalence, showing the percentage of time spent in an oscillatory event of a specific frequency, also referred to as $P_{episode}(f)$.

BOSC algorithm provides logical matrix of the form [number of frequencies * timepoint] where values are 1 when an event was detected and 0 otherwise. For bout duration distributions we summed values across frequency ranges of interest and then performed the logical operation larger than 1, thus in MATLAB it will be, where detected is the BOSC output:

```
thresholded = sum(detected(ThetaRange,:)) ≥ 1;
```

We found start (thresholded switches from 0 to 1) and stop (thresholded switches from 1 to 0) of events and removed events that were incomplete, only had start or stop, and computed duration. We then fitted a kernel probability distribution to the duration values using fitdist function in MATLAB. We used bandwidth of 50 and 10 for theta and gamma oscillations, respectively.

### Spike-field synchronization

To quantify spike-field synchronization, we used fieldtrip toolbox (MATLAB) to compute PPC which is unbiased by the number of spikes (*Vinck et al., 2010*). Raw continuous recordings were resampled with a 1000 Hz sampling rate. The spectral content was estimated with a frequency-dependent Hanning window with 5 cycles per frequency and frequency resolution of 1 Hz. All detected spikes of a unit during the session were included. To assess the statistical significance of spike-field synchronization, we first used a non-parametric permutation test with minimal assumptions. In this procedure, the distribution of PPC values was estimated from 1000 iterations of shuffled spike times of each cell. We used the PPC distribution of shuffles to compute the PPC threshold for significance at each frequency. We applied a threshold of uncorrected $p<0.05$ to determine the significant synchronization at each frequency. Only PPC values that exceeded the statistical threshold and had a Rayleigh test $p<0.05$ and a minimum peak and peak prominence of 0.005 were reported as significant. To obtain the probability distribution of observing significant PPC values at a frequency, we fitted a kernel probability distribution to significant frequency values using fitdist function in MATLAB. We used a bandwidth of 4.

To compare spike-field synchronization during SWRs, we extracted spikes inside a 600 ms window centered around the SWR events and computed PPC [$PPC_{SWR}$]. These spikes were then excluded from the unit spike timestamps and PPC was calculated for the remaining 'residual' spikes [$PPC_{residual}$]. $PPC_{SWR}$ was then compared with $PPC_{residual}$. Only cells with at least 20 spikes during ripple time windows were included in this analysis (N = 185).

To test the significance of differences in spike-field coupling within SWR epochs or excluding them, on a per-unit basis, spikes were randomly selected and assigned to SWR and residual conditions. In this random selection, spike counts were controlled to correspond to the original condition. We performed the random selection 1000 times and measured the difference between PPC in each iteration to obtain the null distribution. Then, we grouped frequencies into six bands 2–3, 4–10, 11–20, 21–40, 41–100, 101–200 Hz. In each frequency band, we found the peak frequency at which the absolute PPC difference was largest and only tested these for significance. If the p-value of PPC difference was less than 0.05 (two-tailed) after FDR correction, it was labeled as significant.

### Theta modulation index estimation

We used the method described by *Royer et al., 2010*, to quantify the degree of theta modulation in single units. For all units, we first computed the autocorrelogram of the cell, in 10 ms bins from –500 ms to +500 ms, normalized to the maximum value between 100 ms and 150 ms (corresponding to theta modulation), and clipped all values above 1. We only included autocorrelograms with at least 100 counts for further steps (*N*=240 units). We then fit each autocorrelogram with the following function:

$$y(t) = \left[ a \left( \sin(\omega t) + 1 \right) + b \right] * e^{-|t|/\tau_1} + c * e^{-t^2/\tau_2^2}$$

where $t$ is the autocorrelogram time lag from –700 ms to 700 ms, and $a - c$, $\omega$, and $\tau_{1-2}$ were fit using the *fminsearch* optimization function in MATLAB. The theta indexes were defined as the ratio of the fit parameters $a/b$. For best-fitting performance, we restricted possible values for $\omega$ to (4, 10), for $a$ and $b$ to non-negative values, for $c$ to (0, 0.2), and for $\tau_2$ to (0, 0.05).

### Additional single-unit datasets

To generate example plots of theta rhythmic cells (*Figure 3*), recordings from the Buzsáki laboratory were included (https://buzsakilab.nyumc.org/datasets/).

## Acknowledgements

The authors wish to thank A Maurer for thoughtful comments on the manuscript and W Zinke and I Hayes for technical support for the wireless recordings.

## Additional information

### Funding

| Funder | Grant reference number | Author |
|---|---|---|
| National Institutes of Neurological Disorders and Stroke | R01NS127128 | Saman Abbaspoor Kari L Hoffman |
| Whitehall Foundation | | Kari L Hoffman |
| Alzheimer's Society of Canada Doctoral Award | | Ahmed T Hussin |
| National Science and Engineering Research Council | Discovery Grant | Ahmed T Hussin Kari L Hoffman |
| NSERC CREATE Vision Science and Applications | | Ahmed T Hussin Kari L Hoffman |
| Brain Canada Multi-Investigator Research Initiative | | Ahmed T Hussin Kari L Hoffman |
| The Krembil Foundation | | Ahmed T Hussin Kari L Hoffman |
| National Eye Institute | P30 EY008126 | Kari L Hoffman Saman Abbaspoor |

The funders had no role in study design, data collection and interpretation, or the decision to submit the work for publication.

### Author contributions

Saman Abbaspoor, Data curation, Formal analysis, Investigation, Visualization, Methodology, Writing - original draft, Writing - review and editing; Ahmed T Hussin, Data curation, Investigation, Methodology; Kari L Hoffman, Conceptualization, Data curation, Supervision, Funding acquisition, Investigation, Visualization, Writing - original draft

### Author ORCIDs

Saman Abbaspoor ⬦ http://orcid.org/0000-0002-3550-0617
Kari L Hoffman ⬦ http://orcid.org/0000-0003-0560-8157

### Ethics

This study was performed in strict accordance with the recommendations in the Guide for the Care and Use of Laboratory Animals of the National Institutes of Health. All of the procedures were in accordance with a protocol approved by the local governing authorities. In the US this was the institutional animal care and use committee (IACUC # M1700152), and in Canada, this was the Canadian Council on Animal Care, local Animal Care Committee at York University (#2014-9).

### Decision letter and Author response

Decision letter https://doi.org/10.7554/eLife.86548.sa1
Author response https://doi.org/10.7554/eLife.86548.sa2

## Additional files

### Supplementary files
• MDAR checklist

## Data availability

The code used to process these data are available at https://github.com/hoffman-lab/Manuscripts/tree/main/AbbaspoorHussinHoffman2023 (copy archived at *Abbaspoor et al., 2023*). Data structures can be downloaded at https://zenodo.org/record/7757458. Previous reports from the stationary data are *Hussin et al., 2020*; *Leonard and Hoffman, 2017*; *Leonard et al., 2015*.

The following dataset was generated:

| Author(s) | Year | Dataset title | Dataset URL | Database and Identifier |
|---|---|---|---|---|
| Abbaspoor S | 2023 | Theta-and gamma-band oscillatory uncoupling in the macaque hippocampus | https://doi.org/10.5281/zenodo.7757458 | Zenodo, 10.5281/zenodo.7757458 |

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
