## [Editor Report]

The rodent hippocampus is one of the main neuroscience models for memory, navigation, and plasticity, and research has suggested important roles for the theta-rhythmic modulation of firing activity and γ oscillations in these processes. This valuable study shows solid evidence of differences between a non-human primate and the rodent hippocampus, in that theta and γ frequencies are segregated by behavioral states, with theta dominating in quiescence and early sleep and β/γ during visual search.

---

## [Decision Letter]

**Decision letter after peer review:**

[Editors’ note: the authors submitted for reconsideration following the decision after peer review. What follows is the decision letter after the first round of review.]

Thank you for submitting the paper "Theta- and γ-band oscillatory uncoupling in the macaque hippocampus" for consideration by *eLife*. Your article has been reviewed by 3 peer reviewers, and the evaluation has been overseen by a Reviewing Editor and a Senior Editor. The following individual involved in the review of your submission has agreed to reveal their identity: Antonio Fernandez-Ruiz (Reviewer #2).

Comments to the Authors:

Although all three reviewers found the paper interesting and valuable, they all three agreed on major concerns with regard to several points: 1) One concern was that the data from stationary monkeys cannot be directly compared to moving rats and that the only freely moving data is an N=1 data point in the Supplement. 2) Other concerns were raised about the data analysis, e.g. the characterization, interpretation, and detection of theta oscillations. If you feel that you could however address point no. 1 and point no. 2, which would require adding new experimental data (or analyzing experimental data that you may have already collected that is not currently included in the manuscript) in case of point no. 1, we could consider a resubmission of a strongly improved manuscript. Since a paper of this kind would contain new data that has not yet been evaluated, it would be treated as a new submission with no guarantees of being sent out for full review or ultimate acceptance. If you decide to go this route, we would recommend to first draft a short (1-2 page) action letter, which would be shared with the reviewers, prior to submitting a new manuscript.

*Reviewer #1 (Recommendations for the authors):*

This is a terse and easy-to-read manuscript, describing an overall lack of theta-dominance in the monkey hippocampus.

It is unfortunate that the most important data – active movements in the monkey – has N=1 and is relegated to the supplement. This weakens the impact of the paper. Furthermore, we don't even know if the electrodes were truly in the hippocampus (no MRIs or histology is shown), or perhaps this one monkey was just a bit unusual. I dislike making such important neuroscience claims based on a case report (this would not be acceptable in human studies, even in intracranial recordings). But I also acknowledge that this is sometimes done in monkey neuroscience.

The paper on its own is certainly thought-provoking, but the authors seem to be unfamiliar with the literature in humans, showing search-, memory-, and movement-related theta oscillations in the hippocampus. I'm not arguing that human hippocampal theta is the same as rodent theta (I don't think it is), but it's not really accurate to make claims about "primate" hippocampal theta based on these findings alone.

The y-axis limits in 2A-B are all very different. Indeed, the PPC magnitudes shown in panel A are basically at the noise level in panel B. That makes panels C-D difficult to interpret, because it's not clear whether the units are robustly (probably meaningfully) modulated by the LFP, or just statistically significantly so (that is, p<.05 but tiny effect size due to large sample size). This is not mentioned or discussed critically.

Perhaps get more data from monkey hippocampus and/or human hippocampus to substantiate these conclusions.

*Reviewer #2 (Recommendations for the authors):*

Abbaspoor and colleagues investigate in this work the interactions between different oscillatory patterns in the macaque hippocampus. This is an important topic that has been profusely investigated in other model species (mostly rodents), but the apparent differences with primates highlight the need for more comparative studies in the field. The present study has been conducted with the highest standard regarding both experimental and analytical methods. The results presented offer strong support for the main conclusion of the authors (the anticorrelation between theta and γ activities, contrary to what happens in rodents). My comments below are aimed at two main aspects of the study that can be further improved: a more comprehensive description of the data and a more extensive comparison with well-known rodent hippocampal physiology. Provided these corrections are incorporated, I believe the present study will add important evidence about macaque hippocampal oscillations and their comparison with those of rodents, thus being of potential interest for a large audience.

– Figure 1 presents a good description of the spectral properties of macaque hippocampal oscillations; however, spatial and temporal aspects of these activities can be better documented. In rodents, theta, γ and ripple oscillations have very stereotyped laminar profiles. Within the limitations of their recording methods, can the authors describe the laminar characteristics of macaque oscillations (e.g. is γ stronger in the apical CA1 dendrites, etc.)? In rodents, γ and ripples typically occur in a short burst while theta can be quasi-stationary for many seconds. What are the temporal dynamics of these oscillations in the macaque?

– As one of the main goals of this work is to compare theta-γ interactions in macaques versus rodents, it will be useful to reproduce the most common analysis used to illustrate this phenomenon in rodents: theta phase – γ amplitude cross-frequency coupling (Tort et al., 2007). This analysis will offer a complementary view to the power-power and bicoherence analysis already presented.

– In lines 78-80 the authors state 20-35Hz oscillations ('slow γ') have been shown in rodents to be the result of theta harmonics. This is a gross mischaracterization of the rodent literature. Surprisingly, the authors have chosen to focus on the only one outlier paper that makes this claim against the overwhelming body of work suggesting the opposite (existence of slow γ independently of theta harmonics). Just to cite some examples: Csicvari et al., Neuron, 2003; Colgin et al., Nature, 2009; Schomburg et al., Neuron, 2014; Fernandez-Ruiz et al., Neuron, 2017; Cabral et al., Neuron, 2014; Lopes dos Santos et al., Neuron, 2018; Lastozcni and Klausberger, Neuron, 2016; Belluscio et al., JNeurosci, 2012. This statement needs to be amended to provide a fair account of the rodent literature or, if that is the position of the authors, much further elaborated.

– In lines 122-125 a spike-SWR analysis is described but SWR has not been described or characterized before this point. Even if it is not the main point of this work, a basic characterization of SWRs needs to be provided. In Figure 1 there are a couple of events that look like ripples, if that is the case, that needs to be mentioned. How did the authors differentiate SWR from fast γ?

– Could the authors identify different γ sub-bands, as is the case in the rodents. If they couldn't, that is also a result worth mentioning, given the potential disparity with rodents.

– The methods pertaining to single unit isolation and classification are not described in the manuscript. How were individual units isolated and classified into cell types? Which cells were used for the spike-LFP analyses? Pyramidal cells and interneurons should be separated for these analyses.

– It needs to be acknowledged in the discussion that some of the reported differences between macaque and rodents could be due to the very different behavioral states in which oscillation has typically been measured in these cases (head-fixed visual search vs freely moving navigation).

– How were sampling periods selected for the analysis of Figure 3? Theta modulation of rodent ACGs will only be clearly present if periods of theta activity (or movement) were selected. How did the authors ensure that the rodent and macaque data are comparable? If no such preselection was done, it needs to be better explained in the results which exact data went into the analysis. A complementary analysis the authors could include in population CCGs, it may be possible that some rhythmic modulation appears in those that are missed in ACGs due to the sparse firing of pyramidal cells. Pyramidal cells and interneurons should not be mixed here.

*Reviewer #3 (Recommendations for the authors):*

The research on hippocampal oscillations has been heavily biased towards rodent models. Comparative studies between rodents and other species are lacking in this field, yet these are vital for the understanding of the mechanisms and functions of these rhythms. Therefore, this study by Abbaspoor, Hussin and Hoffman has to be appreciated for providing descriptive analyses of intracranial recordings from the macaque CA1 and relating them to the widely studied rodent CA1. More specifically, the authors focus on theta and related γ oscillations during rest and a visual search task. They found that CA1 LFPs present stronger 15-70Hz γ and reduced "theta" band power during the visual task compared to rest periods. Spike to phase coupling in the same region indicated that neurons coupled to three bands (3-10, 20-30Hz and 60-150Hz). Importantly, analysis of spike to field coherence indicated that in fact the "theta" power is dominated by irregular (non-oscillatory) activity nesting ripple-like events, i.e. most likely emerging from sharp wave events. The authors point out differences between these observations and what has been reported in the rodent CA1- notably, the lack of theta oscillations during "active" behavior.

Although I appreciate the objective and discussion of the work, the analyses need more refinement to truly support their claims. Further, there are a few apparent conceptual inconsistencies that should be addressed.

My first concern relates to the existence (or not) of true theta oscillations in the data.

The authors indicate in several parts of the manuscript (including the title) that they are analyzing "theta" activity. However, they have not presented convincing indications that they analyzed actual theta oscillations. In fact, spectral peaks do not necessarily demonstrate the presence of a true oscillation, as they write themselves. They find no evidence of genuine theta entrainment of spiking activity, arguing that the apparent theta modulation arises actually from sharp-wave-ripple (SWr) events. It is not clear to me if the authors are claiming the theta band in the LFPs arises solely from SWr in their data. If the authors are actually analyzing theta, they should present evidence this is the case. Although cells are not paced at the theta rhythm, if genuine theta oscillations are present in the LFPs it should be possible to show evidence of that (e.g. as Jutras et al. 2013, Fig1G). In fact, theta bouts/bursts have been reported in primate studies (e.g. Leonard et al. 2015). If the authors are convinced to have theta oscillations in their recordings through the BOSC algorithm, this should be better developed in the manuscript. Further, the authors could analyze changes in theta burst incidence if they want to claim theta is reduced during search, and, most importantly, show that this cannot be explained by a decrease of SWr incidence (which they do show to contaminate the "theta band") relative to rest. One of the main claims of the manuscript is the decoupling between 20-35Hz γ and theta, but this can only be addressed if the authors restrict their analysis to true theta oscillations rather than SWr events. Even by doing so, the authors should discuss the possibility that they do not see γ-to-theta modulation because they record from the pyramidal cell layer whereas γ oscillations have been shown to originate in deeper layers in rodents (e.g. Belluscio et al. 2012; Schomburg et 2014; Lasztóczi and Klausberger 2014). This modulation might be easier to record from the pyramidal cell layer in rodents as γ oscillations have stronger signals in these species due to their anatomical features (Buzsáki et al. 2003).

Alternatively, if the authors cannot show convincingly they are analyzing true theta oscillations, what they actually observe is a reduction (although very mild, as shown in Fig1D) of SWr incidence (reflected in a reduction of "theta" band power) during visual search compared to rest, which would be actually consistent with the rodent literature. What could be argued to be a contrast between species is that SWr rate is much higher in primates compared to rodents during "active behavior", but this has already been claimed by the same group (Leonard et al. 2015; Leonard et al. 2017). Moreover, primate visual search is taken here as analogous to rodent overt movement, which is arguably not a fair equivalence. In fact, SWr events have been shown to occur in rodents during decision making and planning when animals are relatively immobile but arguably attentive (Joo and Frank 2018). Here, the authors have a single freely moving animal (shown in the figure supplements), but lack of constant theta oscillations in similar conditions have been shown before (Ekstrom et al., 2005; Watrous et al., 2013; Talakoub et al., 2019). In fact, different correlates between theta and behavior in different species have been discussed for decades (e.g. Winson 1972). The authors should refine their discussion on the equivalence of "active behavior" between species and rewrite the paper according to the fact they are studying SWr events, and not genuine theta oscillations if that is the case. Also, regarding this point, they should clearly discuss what are the novel aspects of their current findings in relation to the literature.

My second point relates to the relevance of the decoupling between 20-35Hz γ and theta.

The authors highlight the observation that 20-35Hz γ oscillations are not coupled to theta. This has been shown before in primates during visual search (Leonard et al. 2015); and also has been shown in rodents. For example, 23-30 Hz (beta2) oscillations occur in the absence of theta (Berke et al. 2008; Figure 3) in freely moving mice, and γ can be recorded in slow-wave sleep in rats (Isomura et al. 2006 Fig7C). Thus, it is not established that γ oscillations in rodents can only occur coupled to theta. Therefore, the presence of this γ rhythm in the absence of theta reported in the manuscript does not strike me as a relevant contrast between primates and rodents. A contrast would be the absence of continuous theta in the primates, which was extensively reported before. I might have missed something, so I would suggest that the authors make clearer what the novel observations presented here, and how these represent an important contrast between species.

---

## [Author Response]

[Editors’ note: the authors resubmitted a revised version of the paper for consideration. What follows is the authors’ response to the first round of review.]

Comments to the Authors:Although all three reviewers found the paper interesting and valuable, they all three agreed on major concerns with regard to several points: 1) One concern was that the data from stationary monkeys cannot be directly compared to moving rats and that the only freely moving data is an N=1 data point in the Supplement. 2) Other concerns were raised about the data analysis, e.g. the characterization, interpretation, and detection of theta oscillations. If you feel that you could however address point no. 1 and point no. 2, which would require adding new experimental data (or analyzing experimental data that you may have already collected that is not currently included in the manuscript) in case of point no. 1, we could consider a resubmission of a strongly improved manuscript. Since a paper of this kind would contain new data that has not yet been evaluated, it would be treated as a new submission with no guarantees of being sent out for full review or ultimate acceptance. If you decide to go this route, we would recommend to first draft a short (1-2 page) action letter, which would be shared with the reviewers, prior to submitting a new manuscript.

We are pleased that the reviewers appreciated the value of the manuscript, and we took to heart the two main previous concerns raised. 1. As suggested, we have added new, additional data from another freely-moving monkey, demonstrating that the results hold even for the conditions considered more directly comparable to moving rats. 2. As requested, we have added more ‘raw’ data visualization and ran additional analyses, to clarify the (co)occurrence of oscillations, including theta oscillations.

Reviewer #1 (Recommendations for the authors):This is a terse and easy-to-read manuscript, describing an overall lack of theta-dominance in the monkey hippocampus.It is unfortunate that the most important data – active movements in the monkey – has N=1 and is relegated to the supplement. This weakens the impact of the paper. Furthermore, we don't even know if the electrodes were truly in the hippocampus (no MRIs or histology is shown), or perhaps this one monkey was just a bit unusual. I dislike making such important neuroscience claims based on a case report (this would not be acceptable in human studies, even in intracranial recordings). But I also acknowledge that this is sometimes done in monkey neuroscience.

We have added hippocampal CA1 recordings from a third animal (the second freely-behaving monkey) during waking movement and sleep, which showed similar patterns of results to the other monkeys: stronger, more prevalent 2-10Hz activity during sleep compared to stronger, more prevalent 20-80 Hz activity during alert freely-behaving states. Furthermore, theta and γ frequency bands did not show significant coupling. We also include the location of the implanted electrodes in our subjects which targets the CA1 field of the hippocampus (Figure 1 supplement 1) based on combinations of CT/MR coregistration, post-explant MR, and functional markers of the CA1 layer (Figure 2 supplement 1).

The paper on its own is certainly thought-provoking, but the authors seem to be unfamiliar with the literature in humans, showing search-, memory-, and movement-related theta oscillations in the hippocampus. I'm not arguing that human hippocampal theta is the same as rodent theta (I don't think it is), but it's not really accurate to make claims about "primate" hippocampal theta based on these findings alone.

We have endeavored to better articulate our claims. Over the last decade we have recorded in both humans and macaques as they perform similar tasks, giving us a relatively unique perspective on their commonalities and differences. This has required a stationary-subject preparation since the human participants were hospitalized for their medication-resistant seizures, and would have undergone a craniotomy and macroelectrode implantation with tethered cabling shortly before testing, such that mobile recordings were not feasible for them. (The limitations of this preparation, the reviewer may recognize, is in accordance with nearly all other reports of human MTL recordings.) In the present study we capitalize on the advantages of recording from healthy primates, with data obtained from 60 sessions in total across the 3 animals, each with experimentally-targeted and localized microelectrode recordings to one hippocampal subfield layer (Figure 2 supplement 1), across behavioral states including in movement and in stationary hippocampal-dependent search tasks. To better address how these results are situated relative to the human patient literature, we have separated our Discussion section from the results and extended it to discuss the observations in humans as well as other species (275-420). We point to similarities and differences across the main observations and offer potential explanations for some of the discrepancies. We agree with the reviewer -and now state explicitly – that further research will be needed to understand any discrepancies within the primate order. Overall, there were far more commonalities than differences when focusing on the most congruent preparations and methods.

The y-axis limits in 2A-B are all very different. Indeed, the PPC magnitudes shown in panel A are basically at the noise level in panel B. That makes panels C-D difficult to interpret, because it's not clear whether the units are robustly (probably meaningfully) modulated by the LFP, or just statistically significantly so (that is, p<.05 but tiny effect size due to large sample size). This is not mentioned or discussed critically.

Yes, we see various absolute magnitudes of PPC, in line with other PPC results in the literature. To assess whether the significant PPC values are meaningful, i.e. “whether the units are robustly modulated by the LFP”, we carried out the more conservative autocorrelogram fitting analysis. We now discuss the implications of differences across these methods, and offer caution in interpreting PPC phase locking as evidence of oscillatory coupling, when non-stationary signals dominate the LFP (see e.g. 300-308, 318-327, 400-405).

Perhaps get more data from monkey hippocampus and/or human hippocampus to substantiate these conclusions.Reviewer #2 (Recommendations for the authors):Abbaspoor and colleagues investigate in this work the interactions between different oscillatory patterns in the macaque hippocampus. This is an important topic that has been profusely investigated in other model species (mostly rodents), but the apparent differences with primates highlight the need for more comparative studies in the field. The present study has been conducted with the highest standard regarding both experimental and analytical methods. The results presented offer strong support for the main conclusion of the authors (the anticorrelation between theta and γ activities, contrary to what happens in rodents). My comments below are aimed at two main aspects of the study that can be further improved: a more comprehensive description of the data and a more extensive comparison with well-known rodent hippocampal physiology. Provided these corrections are incorporated, I believe the present study will add important evidence about macaque hippocampal oscillations and their comparison with those of rodents, thus being of potential interest for a large audience.– Figure 1 presents a good description of the spectral properties of macaque hippocampal oscillations; however, spatial and temporal aspects of these activities can be better documented. In rodents, theta, γ and ripple oscillations have very stereotyped laminar profiles. Within the limitations of their recording methods, can the authors describe the laminar characteristics of macaque oscillations (e.g. is γ stronger in the apical CA1 dendrites, etc.)? In rodents, γ and ripples typically occur in a short burst while theta can be quasi-stationary for many seconds. What are the temporal dynamics of these oscillations in the macaque?

This is a fair point. We have added a figure showing the duration of the detected epochs of theta and γ band events as a function of behavioral state (Figure 1 supplement 4) and readers can hopefully get a sense of the prevalence and durations from the sample video (Video 1) scrolling through sleep and waking states, with detected bouts highlighted. We only used in-layer recordings here, positioned by microdrive, but agree that laminar profiles of these oscillations in different behavioral states is an important addition to clarify candidate homologies and mechanisms. We plan to address in future studies.

– As one of the main goals of this work is to compare theta-γ interactions in macaques versus rodents, it will be useful to reproduce the most common analysis used to illustrate this phenomenon in rodents: theta phase – γ amplitude cross-frequency coupling (Tort et al., 2007). This analysis will offer a complementary view to the power-power and bicoherence analysis already presented.

We follow this logic in principle, but in practice, we believe the bicoherence measure covers the effects of the Tort method while avoiding its specific danger of false positives. Bicoherence is generally a similar measure of PAC, but with less parameterization (Hyafil, Frontiers in Neuroscience, 2015). Further, bicoherence has a higher frequency resolution compared to other measures of PAC (Giehl et al. Neuroimage, 2021), allowing it to dissociate true from spurious cross-frequency couplings (Kramer et al. 2008).

– In lines 78-80 the authors state 20-35Hz oscillations ('slow γ') have been shown in rodents to be the result of theta harmonics. This is a gross mischaracterization of the rodent literature. Surprisingly, the authors have chosen to focus on the only one outlier paper that makes this claim against the overwhelming body of work suggesting the opposite (existence of slow γ independently of theta harmonics). Just to cite some examples: Csicvari et al., Neuron, 2003; Colgin et al., Nature, 2009; Schomburg et al., Neuron, 2014; Fernandez-Ruiz et al., Neuron, 2017; Cabral et al., Neuron, 2014; Lopes dos Santos et al., Neuron, 2018; Lastozcni and Klausberger, Neuron, 2016; Belluscio et al., JNeurosci, 2012. This statement needs to be amended to provide a fair account of the rodent literature or, if that is the position of the authors, much further elaborated.

We acknowledge that the statement was focused on a narrow critique of the slow γ oscillation that has been rebutted in the literature. We removed the statement from the paper and included many of the mentioned articles in the Discussion section.

– In lines 122-125 a spike-SWR analysis is described but SWR has not been described or characterized before this point. Even if it is not the main point of this work, a basic characterization of SWRs needs to be provided. In Figure 1 there are a couple of events that look like ripples, if that is the case, that needs to be mentioned. How did the authors differentiate SWR from fast γ?

We have added a description of ripple detection in the methods (490-506), and show examples from all 3 animals in Figure 2 supplement 1. We used similar methods to those described in Leonard et al., 2015. There, we used a clustering algorithm to show the difference in SWR detection versus fast γ and HFOs. Ideally, one needs to incorporate dendritic layer recordings for segregation; however, as described in the recent consensus statement on SWRs (Liu et al., 2022), that is not always used. As we begin to incorporate laminar recordings we hope to be able to elucidate the possibility of phase-amplitude coupling that is not due to SWRs, outside the pyramidal layer.

– Could the authors identify different γ sub-bands, as is the case in the rodents. If they couldn't, that is also a result worth mentioning, given the potential disparity with rodents.

We agree that identifying sub bands is an important addition to our understanding of the circuit function of CA1. While we note the division between low and high γ (e.g. Figures 1 and 2), we hesitate to make any strong claims about sub-bands, because those are typically measured from LFP outside the pyramidal layer, where we are recording. As indicated above, our future studies will be better suited to describe this.

– The methods pertaining to single unit isolation and classification are not described in the manuscript. How were individual units isolated and classified into cell types? Which cells were used for the spike-LFP analyses? Pyramidal cells and interneurons should be separated for these analyses.

We have included in the method section the criteria for spike sorting (476-480), which was also described in Leonard et al., 2015 and Hussin et al., 2020. For this study, we didn’t perform functional cell-type classification (as we did in Hussin et al., 2020), to be as inclusive as possible in our pool of cells that might show modulation. We couldn’t identify any strong example of theta rhythmicity in the single examples of autocorrelograms in our cell populations, therefore functional classification of the cell types would not change the results. Furthermore, we performed the same analysis on the rat dataset for a fair comparison and visualization when cells are not segregated by type.

– It needs to be acknowledged in the discussion that some of the reported differences between macaque and rodents could be due to the very different behavioral states in which oscillation has typically been measured in these cases (head-fixed visual search vs freely moving navigation).

Yes! Freely-moving navigation could easily make a difference to data obtained in head-fixed animals. To address this, we included the hippocampal LFP recordings from 2 freely-behaving monkeys during movement. As described in the text (98-110) in both subjects, the pattern of results remained qualitatively the same as in the stationary animal. For example, there was no correlation between theta and γ in these sessions (Figure 1 supplement 3). The power and prevalence of theta activity was much larger during sleep (Video 1, Figure 1 supplement 4), consistent with an even purer measure of the offline state in the overnight sleep recordings. Furthermore, we measured velocity in the video to demonstrate the relative weakness of theta during movement. See also reference to free movement in the discussion (272-274, 385-390).

– How were sampling periods selected for the analysis of Figure 3? Theta modulation of rodent ACGs will only be clearly present if periods of theta activity (or movement) were selected. How did the authors ensure that the rodent and macaque data are comparable? If no such preselection was done, it needs to be better explained in the results which exact data went into the analysis. A complementary analysis the authors could include in population CCGs, it may be possible that some rhythmic modulation appears in those that are missed in ACGs due to the sparse firing of pyramidal cells. Pyramidal cells and interneurons should not be mixed here.

The sampling period was the whole session. To ensure that our methods weren’t unfairly biasing us against seeing periodic spiking, we took the same unit populations (not segregated) from the homologous area of the rat, across the full session. Whereas ACGs as a population still showed strong theta periodicity overall in the rat, we found no evidence of theta periodicity in the monkey (Figure 3C), consistent with other reports mentioned in the text (e.g. Courelis et al., 2019). We discuss this in the context of other non-rodent studies of spiking oscillations (312 – 327).

Reviewer #3 (Recommendations for the authors):The research on hippocampal oscillations has been heavily biased towards rodent models. Comparative studies between rodents and other species are lacking in this field, yet these are vital for the understanding of the mechanisms and functions of these rhythms. Therefore, this study by Abbaspoor, Hussin and Hoffman has to be appreciated for providing descriptive analyses of intracranial recordings from the macaque CA1 and relating them to the widely studied rodent CA1. More specifically, the authors focus on theta and related γ oscillations during rest and a visual search task. They found that CA1 LFPs present stronger 15-70Hz γ and reduced "theta" band power during the visual task compared to rest periods. Spike to phase coupling in the same region indicated that neurons coupled to three bands (3-10, 20-30Hz and 60-150Hz). Importantly, analysis of spike to field coherence indicated that in fact the "theta" power is dominated by irregular (non-oscillatory) activity nesting ripple-like events, i.e. most likely emerging from sharp wave events. The authors point out differences between these observations and what has been reported in the rodent CA1- notably, the lack of theta oscillations during "active" behavior.Although I appreciate the objective and discussion of the work, the analyses need more refinement to truly support their claims. Further, there are a few apparent conceptual inconsistencies that should be addressed.My first concern relates to the existence (or not) of true theta oscillations in the data.The authors indicate in several parts of the manuscript (including the title) that they are analyzing "theta" activity. However, they have not presented convincing indications that they analyzed actual theta oscillations. In fact, spectral peaks do not necessarily demonstrate the presence of a true oscillation, as they write themselves. They find no evidence of genuine theta entrainment of spiking activity, arguing that the apparent theta modulation arises actually from sharp-wave-ripple (SWr) events. It is not clear to me if the authors are claiming the theta band in the LFPs arises solely from SWr in their data. If the authors are actually analyzing theta, they should present evidence this is the case.

This reviewer’s perspective helped us clarify our message -- see the additional Video, Figure 1 supplement 4 and Figure 2 supplement 1, and 308-311. We clearly see epochs containing theta oscillations, which are prevalent in rest and sleep, and not merely contamination by SWRs. Indeed, our results suggest that the acceptance of non-stationary, non-oscillatory activity in primates (e.g. during arousal) may have given a false impression about what oscillations are happening during which behavioral states. We will discuss further below.

Although cells are not paced at the theta rhythm, if genuine theta oscillations are present in the LFPs it should be possible to show evidence of that (e.g. as Jutras et al. 2013, Fig1G). In fact, theta bouts/bursts have been reported in primate studies (e.g. Leonard et al. 2015). If the authors are convinced to have theta oscillations in their recordings through the BOSC algorithm, this should be better developed in the manuscript. Further, the authors could analyze changes in theta burst incidence if they want to claim theta is reduced during search, and, most importantly, show that this cannot be explained by a decrease of SWr incidence (which they do show to contaminate the "theta band") relative to rest. One of the main claims of the manuscript is the decoupling between 20-35Hz γ and theta, but this can only be addressed if the authors restrict their analysis to true theta oscillations rather than SWr events. Even by doing so, the authors should discuss the possibility that they do not see γ-to-theta modulation because they record from the pyramidal cell layer whereas γ oscillations have been shown to originate in deeper layers in rodents (e.g. Belluscio et al. 2012; Schomburg et 2014; Lasztóczi and Klausberger 2014). This modulation might be easier to record from the pyramidal cell layer in rodents as γ oscillations have stronger signals in these species due to their anatomical features (Buzsáki et al. 2003).

We note that slow and fast γ oscillations are detectible in the layer (as indeed, we do see them!). Studies of laminar segregation find differences in relative magnitude and current sources, but the signal is measurable in the layer and indeed, phase amplitude coupling in rats and mice is evident in single LFP traces from the CA1 layer. See Author response image 1, from the cited article above, Belluscio et al., 2012, where the raw signal is taken from the pyramidal layer and the CSD plots show S/M/F γ in the filtered

traces at the Pyr layer (depth = 0). Consider the implication, if we are detecting slow and fast γ bouts in the layer, but that the PAC would happen with some invisible, distal γ not detected with our pyr-layer recordings. That would mean that there was another γ to be explained: whatever it was we were seeing in the layer. We will discuss more below, but appreciate the importance of the comment.

**Author response image 1. sa2fig1:** 

Alternatively, if the authors cannot show convincingly they are analyzing true theta oscillations, what they actually observe is a reduction (although very mild, as shown in Fig1D) of SWr incidence (reflected in a reduction of "theta" band power) during visual search compared to rest, which would be actually consistent with the rodent literature. What could be argued to be a contrast between species is that SWr rate is much higher in primates compared to rodents during "active behavior", but this has already been claimed by the same group (Leonard et al. 2015; Leonard et al. 2017). Moreover, primate visual search is taken here as analogous to rodent overt movement, which is arguably not a fair equivalence. In fact, SWr events have been shown to occur in rodents during decision making and planning when animals are relatively immobile but arguably attentive (Joo and Frank 2018). Here, the authors have a single freely moving animal (shown in the figure supplements), but lack of constant theta oscillations in similar conditions have been shown before (Ekstrom et al., 2005; Watrous et al., 2013; Talakoub et al., 2019). In fact, different correlates between theta and behavior in different species have been discussed for decades (e.g. Winson 1972). The authors should refine their discussion on the equivalence of "active behavior" between species and rewrite the paper according to the fact they are studying SWr events, and not genuine theta oscillations if that is the case. Also, regarding this point, they should clearly discuss what are the novel aspects of their current findings in relation to the literature.

We have made substantial changes. This includes additions to the text, which has a new Discussion section, with the collection of additional data, and by performing additional analyses. The main claims of the paper are: (1) The power and prevalence of theta-band activity is greater during inactive behavioral states (sleep and rest in dark) compared to active behavioral states such as search, and this includes true theta oscillations. We included a Video (Video 1) which shows a recording example of hippocampal LFP during sleep and rest and the decomposition of the signal, and bout detection, demonstrating oscillation in rest not attributable to only SWRs (see also the duration difference in Figure 1 supplement 4). We included BOSC results for head-restrained and freelymoving animals in Figure1 supplement 1 and Figure 1 supplement 2. Thus, we do observe theta oscillations in the hippocampus of NHP but the behavioral correlates are different compared to rodents, and we show how some common analysis methods are prone to contamination. (2) Theta and γ activity are not correlated in NHP hippocampal CA1. We used power-power comodulation and bicoherence analysis to demonstrate this point. Both of these analyses measure the interaction between cross-frequency activity with high resolution, and none points to strong coupling between these two oscillatory modes. If anything, we see negative correlations between these two spectral bands in accordance with the divergence in their prevalent behavioral states. Our final finding we have attempted to clarify: (3) Sharp wave ripples in NHP hippocampus show one concrete example of how common analysis methods may produce inflated measures of low frequency “oscillations”. SWRs can have a strong post-ripple deflection in the LFP, producing spectral energy at <10Hz, which overlaps with the theta oscillations. Further, because SWRs were observed and reported during active states in NHP previously, they can contribute to the power of theta activity, coupling between theta and higher-frequencies, and spike-LFP coupling in the theta range. This does not account for the entirety of our findings, but rather, one contributor. We suggest that future research should keep this type of contribution in mind before making evaluations of theta activity in the primate hippocampus.

My second point relates to the relevance of the decoupling between 20-35Hz γ and theta.The authors highlight the observation that 20-35Hz γ oscillations are not coupled to theta. This has been shown before in primates during visual search (Leonard et al. 2015); and also has been shown in rodents. For example, 23-30 Hz (beta2) oscillations occur in the absence of theta (Berke et al. 2008; Figure 3) in freely moving mice, and γ can be recorded in slow-wave sleep in rats (Isomura et al. 2006 Fig7C). Thus, it is not established that γ oscillations in rodents can only occur coupled to theta. Therefore, the presence of this γ rhythm in the absence of theta reported in the manuscript does not strike me as a relevant contrast between primates and rodents. A contrast would be the absence of continuous theta in the primates, which was extensively reported before. I might have missed something, so I would suggest that the authors make clearer what the novel observations presented here, and how these represent an important contrast between species.

We recognize that there are special cases of γ-theta decoupling in the literature; however, the distinction between existence and prevalence is paramount. It is difficult to overstate the implications of describing an unusual or niche appearance of an oscillation, versus a dominant, reliable feature of a behavioral state that our mechanistic models must account for. Currently, the dual oscillatory regime of theta-γ versus SWR is still being applied across species to describe online/offline states, respectively. If this paper’s results hold true, this conceptualization is wrong in primates, at least in the common cases.